



# Predicting Drought and Subsidence Risks in France

Charpentier Arthur[1], James Molly[2], and Ali Hani[3]

[1]UQAM, Université du Québec à Montréal (UQAM), Montréal (Québec), Canada
[2]EURo Institut d'Actuariat (EURIA), Université de Brest, France
[3]Willis Re, Paris, France

**Correspondence:** Arthur Charpentier (charpentier.arthur@uqam.ca)

**Abstract.** The economic consequences of drought episodes are increasingly important, although they are often difficult to apprehend in part because of the complexity of the underlying mechanisms. In this article, we will study one of the consequences of drought, namely the risk of subsidence (or more specifically clay shrinkage induced subsidence), for which insurance has been mandatory in France for several decades. Using data obtained from several insurers, representing about a quarter of the household insurance market, over the past twenty years, we propose some statistical models to predict the frequency but also the intensity of these droughts, for insurers, showing that climate change will have probably major economic consequences on this risk. But even if we use more advanced models than standard regression-type models (here random forests to capture non linearity and cross effects), it is still difficult to predict the economic cost of subsidence claims, even if all geophysical and climatic information is available.

## 1 Introduction

For the insurance industry, climate change is a challenge since risks are increasing, in terms of frequency and intensity, as discussed in McCullough (2004), Mills (2007), Charpentier (2008) or Schwarze et al. (2011). In this article, we will see if it is possible to predict, for a given year, the costs associated with drought, and more specifically here, clay shrinkage induced subsidence, in France.

### 1.1 Drought and climate change

In a seminal book published twenty years ago, Bradford (2000) started to address the problem of getting a better understanding of the connections between drought and climate change, already suggesting that the frequency and the intensity of such events could increase in the future. And in the more recent book, Iglesias et al. (2019) provided additional evidence about the influence of climate change on meteorological droughts in Europe. Ionita and Nagavciuc (2021) studied the temporal evolution of three drought indices over 120 years (the standardized precipitation index – SPI, the standardized precipitation evapo-transpiration index – SPEI, and the self-calibrated Palmer drought severity index – scPDSI). This updated study regarding the trends and changes in drought frequency in Europe concluded that most of the severe drought events occurred in the last two decades, corresponding to the time after the publication of Lloyd-Hughes and Saunders (2002), for example. Similarly, Spinoni et al. (2015) and Spinoni et al. (2017) (studying more specifically Europe, following their initial, all over the world in Spinoni et al.


(2014)) observed that for both for frequency and severity, the evolution towards drier conditions is more relevant in the last three decades over Central Europe in spring, the Mediterranean area in summer, and Eastern Europe in autumn (using also multiple indices, over 60 years).

Regarding economic impacts, Hagenlocher et al. (2019) provided a state of the art of scientific publications over the past twenty years (presenting the outcomes of a systematic literature review of people-centered drought vulnerability and risk con-

ceptualization and assessments). Naumann et al. (2021) shows that (in Europe), drought damages could strongly increase with global warming and cause a regional imbalance in future drought impacts. They provide some forecasts, under the assumption of absence of climate action (+4°C in 2100 and no adaptation), annual drought losses in the European Union and United Kingdom combined are projected to rise to more than 65 billion € per year compared with 9 billion € per year currently, still two times larger when expressed relative to the size of the economy. Note that this corresponds to the general feeling of

the insurance industry: Bevere and Weigel (2021) suggests that, regarding climate change, drought trends and socio-economic factors, not only will it persist, but even accelerate further.

But at the same time, Naumann et al. (2015) pointed out that related direct and indirect impacts are often difficult to quantify. A key issue is that the lack of sufficient quantitative impact data makes it complicated to construct a robust relationship between the severity of drought events and related damages. Insurance coverage for drought has been intensively studied, when related

to agriculture. Iglesias et al. (2019) mentions some drought insurance schemes, with either indemnity based mechanisms, but also drought index based insurance, in Section 2.8. Vroege et al. (2019) provides an overview of index-based insurances in Europe and North America, in the context of droughts, while Bucheli et al. (2021) focuses on Germany. Note that Tsegai and Kaushik (2019) addresses the importance of designing insurance products which does not only address drought impacts but also minimize land degradation. Besides this theoretical work, some countries provide actual covers for such risks. For example

in Spain, it is possible to insure rain-fed crops against drought, as discussed in Entidad Estatal de Seguros Agrarios (ENESA) (2012), but in most country, drought coverage is only concerned with respect to agricultural (crop) insurance, as such as frost (see also Pérez-Blanco et al. (2017)).

## 1.2  From drought to subsidence

In this article, we will use data from several insurance companies in France, regarding a very specific drought related risk, that

is clay shrinkage induced subsidence (even if we predict only a part of the evaluation of cost assessment of land subsidence, as pointed out in Kok and Costa (2021)). If clay shrinkage induced subsidence is now a well known risk (or at least recognized as a major risk, see for instance Doornkamp (1993), or Brignall et al. (2002) which assessed the potential effects of climate change on clay shrinkage-induced land subsidence), insurance coverage for subsidence is still uncommon. Almost twenty years ago, as indicated in McCullough (2004), while perils related to earth movement were traditionally excluded from most property

policies, several states (in the United States of America) have mandated coverage for some subsidence related claims (with several limitations). And recently, Herrera-García et al. (2021) proved that subsidence permanently reduces aquifer-system storage capacity, causes earth fissures, damages buildings and civil infrastructure, and increases flood susceptibility and risk.



From an insurer's perspective, Wües et al. (2011) pointed out that as incidents of soil subsidence increase in frequency and severity with climate change, there is a need for systematic managing of the risks through a combination of loss prevention and
risk transfer initiatives (such as insurance).

In France, subsidence is a phenomenon covered by all private property insurance covers, and that enters the scope of the government backed French natural catastrophe regime provided by the *Caisse Centrale de Réassurance* (CCR). It is the second most important peril in terms of costs that the system covers (the first being floods, see Charpentier et al. (2021) for a recent discussion about flood events in France, in the context of climate change). Subsidence risk is defined (Ministère de la transition
écologique et solidaire (MTES) (2016)) as the displacement of the ground surface due to shrinkage and swelling of clayey soils. It is due mainly to the presence of clay in the soil which swells in humid conditions and shrinks in dry ones, thus creating instabilities in the terrain under constructions causing cracks to appear on the floor and walls which can jeopardise the solidity of the building. France, having a temperate climate, has saturated clayey soils, making subsidence predominant during droughts.

However, the past few years have seen this risk exacerbated by the extreme heat waves and lack of rainfall in France (see Caisse Centrale de Réassurance (CCR) (2019)), causing more and more subsidence claims, with little hope of this tendency stopping, given the current climate change context. Indeed, since 1989, 38% of the total costs of claims are concentrated over the period 2015-2019, that is 15% of the total time that subsidence coverage has been in place (as discussed in Mission des Risques Naturels (MRN) (2021)). Furthermore, Soubeyroux et al. (2011) shows that the frequency and intensity of heat waves
and droughts will inevitably increase in the coming century in continental France and new areas that so far have been protected from drought will be at risk. Additionally, AFFA [2015] predicts that the cost of geotechnical droughts will nearly triple in 2040. More recently, the French Geological Survey (BRGM, *Bureau de Recherches Géologiques et Minières*) published a study, Gourdier and Plat (2018), describing extreme historical subsidence events as well as forecasts using various climate change scenarios. It found that the first third of the century will suffer from unusual droughts in both their intensity and spatial
expansion: one in three summers between 2020 and 2050 and one in two summers between 2050 and 2080 are to be as extreme as the summer of 2003 in continental France (the worst subsidence event ever registered by the CCR, see Corti et al. (2009) that focused on the 2003 heatwave in France). Looking at the most pessimistic scenario, a 2003 type event might occur half of the time between 2020 and 2050. One should recall that in 2003, the heat wave caused damage due to the shrinking and swelling of clay, for which compensation (via the natural disaster insurance scheme, and then via an exceptional compensation
procedure for rejected cases) was estimated at approximately 1.3 billion € in Frécon and Keller (2009), while over the period 1989-2002, the average annual cost of geotechnical drought for the natural disaster insurance scheme was more than five times smaller, with 205 million €.

Furthermore, subsidence is a risk with a long declaration period, with on average 80% of the number of claims declared two years after the event. This delay is due to the lengthy acceptance process of subsidence natural catastrophe declarations,
upon which the validity of most claims is dependant. Although most insurers are reinsured against this peril with the CCR, the retention rate remains high (50%), it is thus necessary for insurers to develop their own view of this risk in order to estimate





their exposure to this growing hazard. However, the inherent characteristics of subsidence make it a risk that is complex to model: it has slow kinetics and an absence of precise temporal definition, making subsidence models sparse on the market.

Bevere and Weigel (2021) mentioned since 2016, annual inflation-adjusted insured losses have continuously exceeded 600 million €, with an average annual loss close to 850 million , which corresponds to around 50% of the CatNat premiums collected, and makes subsidence possibly one of the most costly natural risk in France. Overall, a high to medium clay shrink-swell hazard affects one-fifth of metropolitan France's soils and 4 million individual houses, as mentioned in Antoni et al. (2017). And Soyka (2021) mentioned that this is not just in France, increased subsidence hazard gains attention in other countries, too (even if this article focuses solely on France).

## 1.3 Agenda

The purpose of this study is to provide a regression-based model that will allow to predict annual frequency and severity of subsidence claims to be made, based on market data and climatic indicators. The model created in this study bases future predictions on past occurrences, thus a historical insurance database was necessary to calibrate the models, alongside indicators of the severity of historical events that could be reproduced into the future. These indicators were created using climatic and geological data that capture the specifics of past events and regional information. The creation of this database and the choice of indicators will be described in section 2. Using this historical data, various models are implemented, chosen to adapt to the particularities of the data, in order to improve the precision of the predictions. There will be three layers to model the costs of subsidence claims (1) a drought event should be officially recognised (corresponding to a binary model, specificities of the French insurance scheme will be discussed in the next section) (2) if there is a drought, the frequency is considered (corresponding to a counting model, classically a Poisson model) (3) for each claim the severity is studied (corresponding to a cost model, here some Gamma model). As we will see, using so-called zero-inflated models, the first two models can be considered simultaneously. Various tree based models were also tested in an attempt to obtain more realistic predictions. In section 3, we will present those models, and we will analyse predictions obtained on the frequency, and more specifically discuss the geographical component of the prediction errors. And finally, in section 4, we will present some models to predict to total costs of subsidence events, in France, and again, study mode carefully the prediction errors, in 2017 and 2018.

## 2 Subsidence risk in France and our dataset

A yearly claim and exposure dataset, sourced from several different French insurers, relative to Multi-Peril Housing Insurance (*Multi-Risque Habitation*) for individual houses in metropolitan France, for the period 2001 to 2018, was used here for frequency and severity of past events. This dataset was enriched with additional information based on geophysical indices usually used to model droughts.



Our dataset was aggregated at the town level[1], with no information about the particularities of each individual contract that could influence the claims (such as number of stories, size, orientation, etc.), creating the need for additional information about the inherent risks of the town and its climatic and geophysics exposure. One of the main issues when modelling subsidence is the absence of precise temporal and geographical definition of a subsidence events. Thus, to combat these issues, indicators

must be created to grasp the geographical and temporal characteristics of past events.

In Section 2.1, after describing briefly the specificity of the French insurance scheme, we will explain the variables of interest we will model afterwards, namely the occurrence of a natural disaster (based on official data in France), the number of houses and building claiming a loss, and the amount of the losses (those last two based on data from three important insurance companies in France, representing about 20% of the French market). Then, in Section 2.2 and 2.3, we will describe possible

explanatory variables (for the occurrence of a disaster, the percentage of houses claiming a loss and the severity). And finally, in Section 2.4 we discuss the use of other variables, mainly geophysical information since we want to predict subsidence, and not droughts in general.

## 2.1   Specificity of the French *Catnat* system

The French "*Régime d'Indemnisation des Catastrophes Naturelles*" (also called the "CatNat regime") started in 1982 (see

Charpentier et al. (2021) for an historical perspective), even if drought damages were added to the (informal) list of perils in 1989, as explained in Magnan (1995) or more recently Bidan and Cohignac (2017). The main idea of the mechanism is that any property damage insurance contract, for individuals as well as for companies, includes mandatory coverage for natural disasters. The assets concerned are buildings used for residential or professional purposes and their furniture, equipment, including livestock and crops, and finally motor vehicles. These assets are insured by multi-risk home insurance, multi-risk

business insurance and motor vehicle insurance. The contractual guarantees (storm, hail and snow, fire...) are also very often attached to household and business contracts. Livestock outside the barn and unharvested crops, on the other hand, are covered differently. It should be stressed here that the respective scope of insurable and non-insurable risks is not defined by law, but is established by case law. Indeed, the natural disaster insurance system is said to be "à péril non dénommé" (or unnamed peril) in the sense that there is no exhaustive list of all risks that are covered. The effects of natural disasters are legally defined in France

as "*uninsurable direct material damage caused by the abnormal intensity of a natural agent, when the usual measures to be taken to prevent such damage could not prevent their occurrence or could not be taken*" (article L.125-1 of the insurance legal code). In practice (and it will be very important in our study) the state of natural disaster is established by an interministerial order signed by the Ministries of the Interior and of Economic Affairs. This order is based on the opinion of an interministerial commission. This commission analyzes the phenomenon on the basis of scientific reports and thus establishes jurisprudence

regarding the threshold of insurability of natural risks. More specifically, in the context of our study regarding drought, requests for recognition of the state of natural disaster are examined for damage caused by differential land movements due to drought

---

[1] Here, we use here the word "*town*" to designate a "*commune*" (in French), or "*municipality*". There are 37,613 towns in metropolitan France (as characterized by their INSEE-"*commune*" code).





and soil rehydration. And a list of towns that have been declared a natural disaster are listed and associated guaranties are then applied, both for individual and commercial policyholders, by (private) insurance companies.

In comparison to other natural catastrophes, subsidence has certain particularities. From an insurance point of view, the
typical event-based definitions of a natural catastrophe, that is possible for cyclones or avalanches, does not apply. Indeed, it has slow kinetics, making it difficult to determine direct links of causality between the event and the claims. Damage can be caused long after the dry periods. However, that link of causality is the very definition of natural catastrophe recognition which makes the implementation of different criteria to determine the causality link of the event essential.

Subsidence was first observed, as a major risk, in France after the drought of 1976 which caused important damage to
buildings. After a similar event in 1989, subsidence was integrated into the French natural catastrophe regime, in the sense that policyholders can claim a loss to their insurance companies. According to Mission des Risques Naturels (MRN) (2019), between 1989 and 2018, more than 11,300 towns have requested natural catastrophe recognition for subsidence and over 9,500 were granted it. The natural catastrophe declarations are published on average 18 months post-event, instead of 50 days for other natural catastrophes and the duration of an event is on average of 50 days for subsidence and of 5 days for other perils
(like floods, avalanches or landslides, among many others). The total cost of subsidence losses reached 11 Billion € mid-2018, which is roughly 16,300€ per claim. The number of towns that have had their request declined has increased since 2003. Overall, the proportion of acceptance is of 61%, however, just taking years subsequent to 2003, the proportion is of only 50%. As mentioned earlier, this might be explained by the fact that, according to the law, those events should be caused by "*the abnormal intensity of a natural agent*". Thus, if a town is claiming losses every years, it ceases to be "*abnormal*", and claims
might be rejected then.

The evolution of the number of natural catastrophes since 1989, which is the year of the oldest natural catastrophe in the dataset, shows that since the 1990s, the number of orders has been quite variable, however, 4 years seem to be abnormally hardly-hit: 2003, 2005, 2011 and 2018. Note finally that Wües et al. (2011) claims that, since subsidence and drought are related to temperature, climate change will increase frequency and intensity of drought.

## 2.2 General considerations regarding drought indices

There exists many different options in terms of drought indicators to characterise their severity, location, duration and timing (see Svoboda and Fuchs (2016) for some exhaustive descriptions). The impact of droughts can vary, depending on the specificities of each drought, captured differently by each indicator. It is thus important to select an indicators with its application in mind. However, the availability of data also plays an important role in the selection, as it limits which indicator can be
constructed.

The criteria to characterise the severity of shrinkage-swelling episodes evolved in 2018, as the old criteria were outdated and overly technical, making them difficult to interpret and explain to the public, see Ministère de l'intérieur (MI) (2019). The new system is based on two factors:



– **A geotechnical factor** pertaining to the presence of clay at risk of swell-shrink phenomenon, in place since 1989. This criterion enables the identification of soils with a predisposition to the phenomenon of shrinkage-swelling depending on the degree of humidity. The analysis is based on technical data established by the *Bureau des Recherche Géologiques et Minières* (BRMG), see Ministère de la transition écologique et solidaire (MTES) (2016). Areas of low, medium and high risk are considered to determine whether the communal territory covered by sensitive soils (medium and high-risk areas) is greater than 3%. This will be discussed in Section 2.4. However, the intensity of shrinkage-swelling is not only due to the characteristics of the soil but also to the weather.

– **A meteorological criterion** defined as a hydro-meteorological variable giving the level of humidity in superficial soils (1m of depth) at 8-kilometre precision level. This variable establishes the humidity of the soil for each season at a communal level called the Soil Water Index (SWI) which varies between 0 and 1, where 0 is a very dry soil and 1 is a very wet soil. A humidity indicator is calculated for every month based on the average of the indicators of the three previous months. This will be discussed in Section 2.2. For example, the indicator for July is fixed using the mean of the indexes for May, June and July, thus considering the slow kinetics of the drought phenomenon that can appear over a few months.

To determine whether a drought episode is considered abnormal, the SWI established for a given month is compared to the indicators for that same month over the previous 50 years. It is considered "*abnormal*" if the indicator presents a return period greater or equal to 25 years, as explained in Ministère de l'intérieur (MI) (2019). It should be stressed here that this return period is defined locally, and not nationally. If one of the months of a season meets the above criteria, in a specific area, then the whole season is eligible to a natural catastrophe declaration for the whole town. If the natural catastrophe criteria are met and the Inter-Ministerial commission declares a subsidence natural catastrophe for the town, if the claims are in direct link with the event and the goods were insured with a property and casualty insurance policy then it will be covered by standard insurance policies. The presence of a threshold set at a 25-year return period indicates that over time, if a commune is regularly hit by extreme droughts, those events will be less and less likely to be declared natural catastrophes as they will lose their exceptional character.

However, over the years, the subsidence criteria have been changed and updated many times,to consider the new kinds of droughts that arose. The first main modification was in 2000, when a criterion based on the hydrological assessment of the soil was added to the criterion assessing the presence of clay in the soil which was the sole criteria before. However, 2003 was hit by an extreme and unusual drought limited only to Summer which was not captured by the criteria in place. Indeed, the uses of the criteria in place at the time would have led to most of the towns requesting a declaration to be refused. Thus, a new criterion was created specifically for 2003. In 2004, the criteria were updated once again, to consider droughts like that of 2003. However, in 2009 a new indicator was applied based on three seasonal Soil Water Indexes (SWI) (Winter, Spring and Summer) as well as the presence of clay in the soils. Finally, in 2018 the SWI criteria were updated to simplify the thresholds and create four seasonal indicators as presented previously and the shrinkage and swelling of clay exposure map was also updated.





## 2.3 Drought indices used as covariates

These indicators can be classed into three big families : The first are meteorological indicators, of which the most common amongst drought-related studies are the Standardised Precipitation Index (SPI), the Palmer Drought Severity Index (PDSI) and the Standardised Precipitation and Evapotranspiration Index (SPEI). These indices are based on precipitation data, as well as temperature and available water content for the two last ones. The SPI is the most widely used because it requires little data (only monthly precipitations needed) and is comparable in all climate regimes (see Kchouk et al. (2021) for a recent review of drought indices). However, those indexes do not capture drought through soil moisture, whereas other indexes such Agricultural and Soil Moisture indicators do. Some common indicators from that family are the Normalised Difference Vegetation Index (NDVI), the Leaf Area Index (LAI) or the Soil Water Storage (SWS) which require more complex data such as spectral reflectance, leaf and ground area, soil type, available water content and more. Finally, the last smaller family of indicators are hydrological indexes, some examples of which are the Streamflow Drought Index (SDI), Standardised Runoff Index (SRI) or the Standardised Soil Water Index (SSWI) - which is used by Météo-France to characterise droughts and applied in the characterisation of soil dryness and climate change in (Soubeyroux et al. (2012)) - which require streamflow values, runoff information or soil water data.

Only a small portion of the above indicators could be considered given the limited data available. Indeed, the data used was the monthly average water content, the monthly average soil temperature and the monthly average daily precipitations at a 9km grid resolution globally, from the *ERA-5 Land* monthly database (Climate Change Service Climate Data Store (CDS) (2020)), between January 1981 and July 2020. Therefore, only the SSWI and SPI were selected as they only require soil water content and precipitation respectively and are simple to implement. Note that this SSWI, recently discussed in Torelló-Sentelles and Franzke (2021), was inspired by Hao and AghaKouchak (2014) and Farahmand and AghaKouchak (2015).

The use of precipitation data alone is the greatest strength of SPI, as it makes it very easy to use and calculate. The ability to calculate over multiple timescales also allows SPI to have a wide scope. There are many articles related to SPI available in the scientific literature, which gives novice users a wealth of resources they can count on for help. Unfortunately, with precipitation as the only input, SPI is deficient when it comes to taking into account the temperature component, which is important for the overall water balance and water use of a region. This drawback can make it more difficult to compare events with similar SPI values, as highlighted in Svoboda and Fuchs (2016).

Similarly, the sole use of soil moisture, makes it a simple indicator to use, but also a deficient one. However, as pointed out in Soubeyroux et al. (2012), the SPI and SSWI are complementary : although they do have similarities, they show some great difference, for example they show that the droughts of 2003 are not linked to an extreme precipitation deficit so can not be measured solely by the SPI. In order to add an extra layer of detail to better characterise droughts, a third indicator was created, based on the soil temperature available, inspired by the same methodology as the SPI. In the rest of this article, it will be referred to as the standardised soil temperature index (SSTI).

To obtain indexes that are comparable all over France, the data must be transformed. Indeed, in its raw state, a dry period is difficult to distinguish amongst a simply dry climate: the same magnitude of low precipitation in areas with very dry climates




will have a very different impact on the soil and on subsidence claims than in wet climates. The SPI methodology McKee et al. (1993) allows the creation of normalised indicators (see also Guttman (1998) for an historical discussion). It is calculated using 3-month cumulative precipitation probabilities by calibrating a gamma distribution to the data, which is then transformed into a standard normal distribution. Thus, it allows the quantification of the seasonal deviation of precipitation compared to

the historical mean. A three month step was chosen as it reflects short- and medium -term drought conditions and provides a seasonal estimation.

In order to obtain indexes with the available data mentioned previously, the same methodology - which is the SPI computation methodology - was applied to the monthly average soil wetness, the monthly average soil temperature and the monthly average daily precipitation separately, thus yielding the three-month SPI, SSWI and SSTI.

The aim of this study is to predict claims at a yearly time scale, creating the need for a yearly indicator. The 12-month sliding SSWI, SPI or SSTI were not chosen as they would not capture the seasonality of droughts. Instead, to obtain yearly indicators, the extremes are taken over the 4 yearly seasonal indicators, with the following formulas[2]:

$$\text{ESSWI}_{z,t} = \min_{s \in \mathcal{S}} \left( \text{SSWI}_{s,z,t} \right), \tag{1}$$

$$\text{ESSTI}_{z,t} = \max_{s \in \mathcal{S}} \left( \text{SSTI}_{s,z,t} \right), \tag{2}$$

$$\text{ESPI}_{z,t} = \min_{s \in \mathcal{S}} \left( \text{SPI}_{s,z,t} \right) \tag{3}$$

where $\mathcal{S}$ denotes the set of seasons, $\mathcal{S} = \{\text{Spring}, \text{Summer}, \text{Autumn}, \text{Winter}\}$, $t \in \{1981, 1982, \cdots, 2020\}$ and $z$ denotes the given location. The ESPI is the Extreme Standardised Precipitation Index, the ESSTI is the Extreme Standardised Soil Temperature Index and the ESSWI is the Extreme Standardised Soil Water Index. This methodology - to our knowledge - has not been applied in the past for the creation of the soil temperature and precipitation indexes in the context of subsidence claim

prediction.

## 2.4  Other spatial explanatory variables

The concentration and presence of clay in the topsoil plays an important role in the occurrence of subsidence as it is caused by the shrinkage and swelling of these particular kinds of soil. Thus, to express the risk of an area, a map was obtained from the Land Use and Cover Area Statistical Survey (LUCAS) database published by the European Soil Data Centre (ESDAC)

(2015), which collects harmonised data about the state of land use and cover over the European Union. This map gives the concentration of clay in the topsoils (soils at 0-20cm depth) and is available at a 500m grid resolution over the European Union.

The soil clay concentration was then aggregated by town keeping the highest concentration of clay in the town, which gives the maps in figure 2a. Other aggregation functions were considered (such as the average), but they were less predictive than the maximum. This map shows that the areas with the highest clay concentrations appear to be in the North-East of France, in

Charente-Maritime (West), around Toulouse (South-West) and along the Mediterranean Sea (South-East).

---

[2]Here, we use either the minimum or the maximum, depending if drought events are related either to low or high values of seasonal indices.



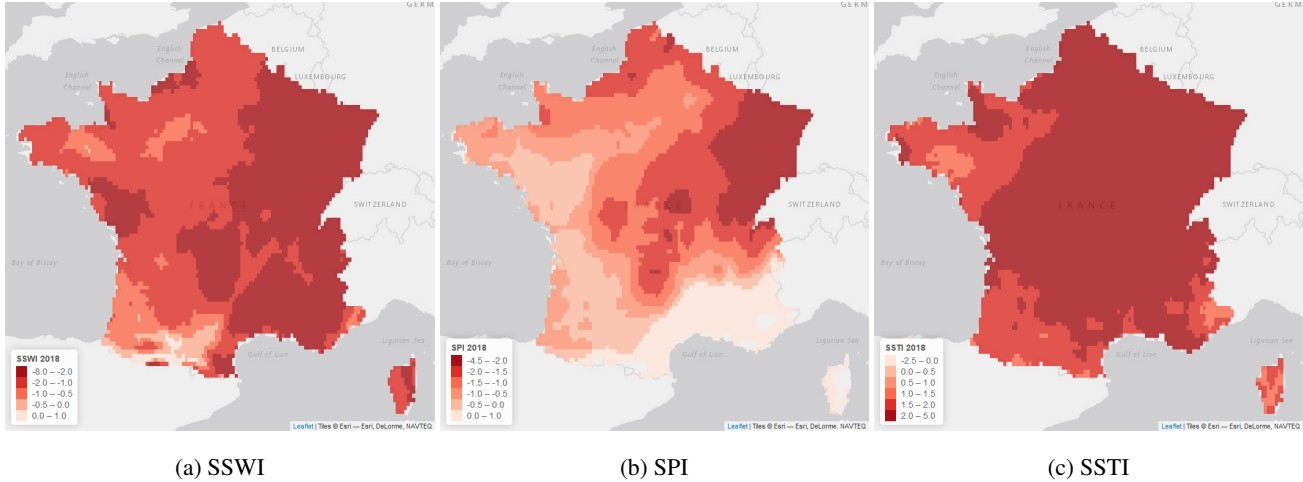

|         |         |         |
|---------|---------|---------|
| (a) SSWI | (b) SPI | (c) SSTI |

**Figure 1.** Indicators for 2018, with SSWI (Standardised Soil Water Index), SPI (Standardised Precipitation Index) and SSTI (Standardised Soil Temperature Index) from left to right.

Finally, a binary categorical variable, which takes the values 1 if the town has historically made a request for a natural catastrophe declaration and 0 otherwise, was sourced from CCRs' historical data, CCR [2020]. The risk map obtained is visible in figure 2b, as at 2018.

These variables were then aggregated to the exposure dataset. Thus, the calibration dataset was composed for each claim year of the "INSEE code" of the town (from the official classification, that can be seen as the ZIP code used in the U.S.), the year of the claim, the number of claims, the cost of claims (in €), the number of policies, the total sums insured (in €), the clay concentration in the soil, the ESPI, the ESSTI, the ESSWI and Cat (the binary categorical variable giving the occurrence of a historical natural catastrophe declaration request, prior to the year of study). All the models calibrated are based on those five variables mentioned above.

## 3 Model calibration for the frequency

Using simulations at regional scale, Corti et al. (2009, 2011) pointed out that it is possible to use simulation to get a good representation of the regions affected by drought-induced soil subsidence, but substantial differences between simulated and observed damages in some regions. In this section, we will describe a simple spatio-temporal model for resilience, to model either the frequency or the intensity of such events (on a yearly basis). Regression type models will be considered as in Blauhut et al. (2016). The main difference is that the interest was to model occurrence, and a logistic regression was sufficient. Here, since we focus on frequency and intensity, some Poisson-type regression models will we considered, first, and then some Gamma models, for the severity, to predict the economic cost of subsidence. In Section 3.3 we will present regression based models, that we will extend in Section 3.4 to ensemble models, namely with bagging of regression trees, as suggested in





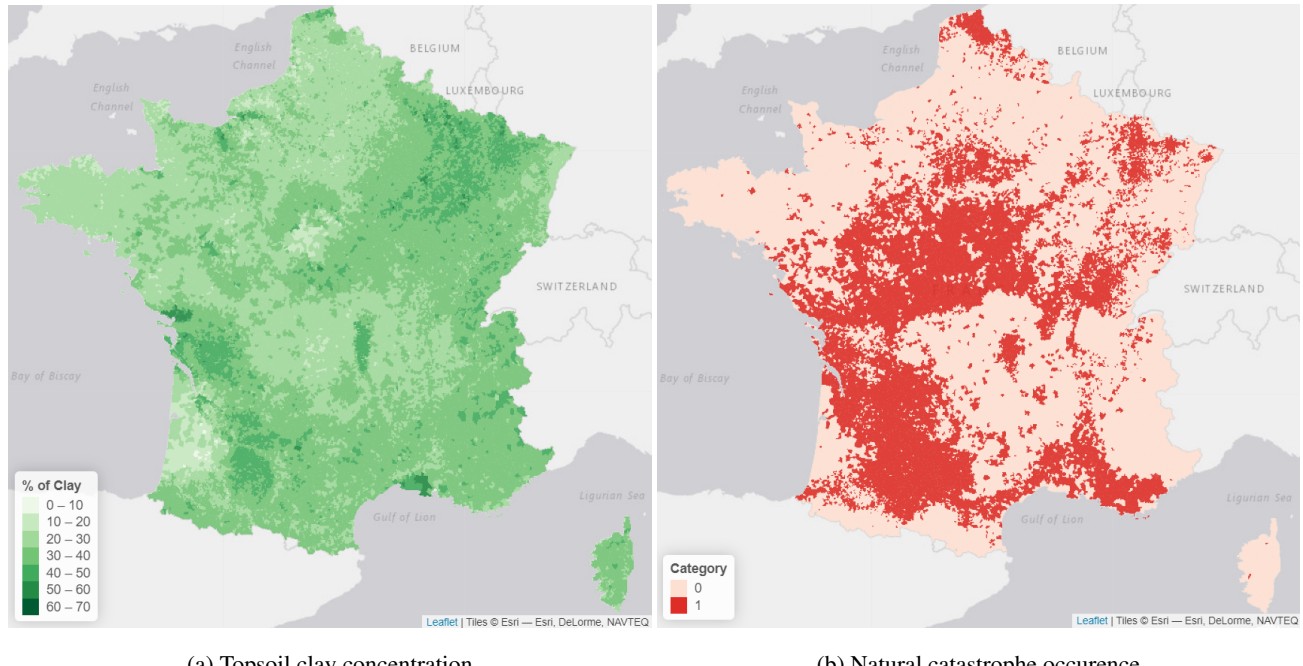

(a) Topsoil clay concentration                    (b) Natural catastrophe occurence

**Figure 2.** Topsoil clay concentration and historical natural catastrophes accepted requests. Those two variables will be used for our predictive model.

Breiman (1996) to take into account some possible non-linearity, as well as cross-effects. And in section 3.5, we will study
more carefully the errors. But before, we need to introduce some criteria to select an appropriate model.

## 3.1    Model selection criteria

Various validation performance measures are used to compare the different models. They are chosen to optimise the model selection based on the qualities that are desired from the model which are: a good capacity at predicting the correct number of claims in the right areas, the ability of not predicting claims where there are none historically all the while keeping the simplest
possible model. The following performance measures are thus used:

– Akaike Information Criteria (AIC) and Bayesian Information Criteria (BIC), which use maximum likelihood and penalises models with respectively too many variables and too many variables with respect to the number of observations. These criteria are only used for the parametric regression models.

– Root Mean Square Error (RMSE) – or simply the sum of the squares of errors, which penalises greatly extreme errors,
i.e. extreme deviations in predictions. This will be used for costs, and not counts.





## 3.2 Cross-Validation method

In order to assess the predictive power of the calibrated models, cross-validation was used. The main goal of the models is to predict accurately the number of claims, per town, on a yearly basis, thus a yearly cross-validation, that is derived from the basic cross-validation principle, will be used.

The idea of cross validation is related to in-sample against out-of-sample testing. In a nutshell, the model is fitted on a subset of the data, and validation is computed on observations that were left out. This approach is interesting to assess how the results of some statistical analysis will generalize to an 'independent' data set. Note that it is possible to remove one single observation, and to validate the model on that one – this is the *leave-one-out* approach – or to split the original dataset on $k$ subgroups, to remove one subgroup to fit a model, and predict on that specific subgroup, and rotate – this is the *k-fold cross-validation*

approach. In the case of spatio-temporal data,

  – use some regions, used as subgroups, then use some spatial $k$-fold approach (as in Pohjankukka et al. (2017)) where the model is fitted on $k-1$ regions, and validation is based using some metric on the error on the region that was left out – it can be the sum of the squares of the difference, or any metric discussed in the previous section,

  – use cross-validation in time, but because of particular properties of the *time* dimension, cross-validation is performed by

removing the future from the analysis (as in Bergmeir et al. (2018)): at time $t$, we use observations up to time $t$ to fit a model, then get a prediction for time $t+1$. In some sense, it is a classical leave-one-out procedure, except that we cannot use observations *after* time $t+1$ to get a prediction at time $t+1$.

## 3.3 Regression-based models

A first attempt at modelling the yearly number of claims was made using Generalised Linear Models (GLM) with Poisson,

Binomial and Negative Binomial distributions, which are the most adapted to the calibration data, on counts, as a starting point. These models offer a simple and interpretable approach to model data by assuming that the response variable $\boldsymbol{Y} = (Y_{z,t}) \in \mathbb{R}^{n \times T}$ is generated by a given distribution and that its mean is linked to the $q$ explanatory variables $\boldsymbol{X} = (\boldsymbol{X}_{z,t}) \in \mathcal{X}^{n \times T}$ (where classically $\mathcal{X} = \mathbb{R}^q$), through a link function. In this model, we have $n$ spatial locations (the number of towns), $T$ dates (the number of years), and $q$ possible explanatory variables.

If we want to model the number of houses claiming a loss in a given location, the Poisson GLM is defined as $Y_{z,t} \sim \mathcal{P}(E_{z,t} \cdot \lambda_{z,t})$, for a location $z$ and a year $t$, where $E_{z,t}$ is the exposure (the number of contracts in the town) and $\lambda_{z,t}$ is the yearly intensity, per house,

$$\lambda_{z,t} = \exp\left[\beta_0 + \beta_1 x_{1,z,t} + \cdots + \beta_k x_{q,z,t}\right] \tag{4}$$

where $x_{j,z,t}$'s are features used for modeling, such as the ESSWI, the ESSTI, etc. The prediction, performed at year $t+1 \in$

$\{2001,...,2019\}$ based on a calibration set of years $\{2001,...,t\}$, is

$$_t\widehat{N}_{z,t+1} = E_{x,t+1} \cdot {}_t\widehat{\lambda}_{x,t+1} \tag{5}$$





and, based on estimator $\widehat{\boldsymbol{\beta}}_t$ obtained on the training dataset with observations of years $\{2001, ..., t\}$, using maximum-likelihood techniques, and

$$
{}_t\widehat{\lambda}_{z,t+1} = \exp\left[\widehat{\beta}_{0,t} + \widehat{\beta}_{1,t}x_{1,z,t+1} + \cdots + \widehat{\beta}_{k,t}x_{q,z,t+1}\right]
\tag{6}
$$

where geophysics covariates $x_{j,z,t+1}$ are known. In Table 1 several sets of parameters estimates $\widehat{\boldsymbol{\beta}}_t$ are given (with $t$ varying from 2008 until 2018).

**Table 1.** Evolution of the parameters in the regression (Poisson regression). Numbers in brackets are the standard deviations of the parameters. Note that two ESSPI parameters are not significant here (95% level).

| year | $t$ | 2008 | 2010 | 2012 | 2014 | 2016 | 2018 |
|---|---|---|---|---|---|---|---|
| Intercept | $\widehat{\beta}_{0,t}$ | -13.668 | -13.460 | -13.522 | -13.735 | -13.932 | -14.357 |
| | | (0.074) | ( 0.071) | ( 0.062) | ( 0.06) | ( 0.059) | ( 0.049) |
| ESSTI | $\widehat{\beta}_{1,t}$ | 1.522 | 1.420 | 1.511 | 1.494 | 1.539 | 1.661 |
| | | (0.017) | ( 0.015) | ( 0.013) | ( 0.013) | ( 0.013) | ( 0.012) |
| ESSWI | $\widehat{\beta}_{2,t}$ | -0.711 | -0.700 | -0.601 | -0.709 | -0.750 | -0.707 |
| | | (0.011) | ( 0.011) | ( 0.009) | ( 0.009) | ( 0.009) | ( 0.008) |
| clay | $\widehat{\beta}_{3,t}$ | 0.021 | 0.020 | 0.024 | 0.025 | 0.025 | 0.035 |
| | | (0.001) | ( 0.001) | ( 0.001) | ( 0.001) | ( 0.001) | ( 0.001) |
| cat | $\widehat{\beta}_{4,t}$ | 3.924 | 3.950 | 3.957 | 3.957 | 4.003 | 3.902 |
| | | (0.056) | ( 0.055) | ( 0.049) | ( 0.048) | ( 0.047) | ( 0.038) |
| ESSPI | $\widehat{\beta}_{5,t}$ | -0.046 | -0.010 | 0.016 | 0.074 | 0.127 | -0.048 |
| | | (0.013) | ( 0.011) | ( 0.009) | ( 0.009) | ( 0.009) | ( 0.007) |

From Table 1, we can see that the model is rather stable over time, which is an interesting feature from a modeler's perspective: if we predict more claims related due to subsidence, it is mainly coming from the underlying factors than from a change in the impact of each variable. It can be mentioned that $\widehat{\beta}_{3,t}$ (associate the clay) is significantly increasing (with a $p$-value of 350  2%).

This Poisson model is rather classical to model counts, and it is said to be an equi-dispersed model, in the sense that the variance of $Y$ is equal to the average value. This model can be extended in two directions, instead of having

- the binomial model, where $Y_{z,t} \sim B(E_{z,t}, p_{z,t})$, where $E_{z,t}$ is the exposure and $p_{z,t}$ is the probability that, for a given year $t$ and location $z$, a claim is made for a single house, and the prediction for $p_{z,t+1}$ is

$$
{}_t\widehat{p}_{z,t+1} = \frac{\exp\left(\widehat{\beta}_{0,t} + \widehat{\beta}_{1,t}x_{1,x,t+1} + \cdots + \widehat{\beta}_{k,t}x_{q,x,t+1}\right)}{1 + \exp\left(\widehat{\beta}_{0,t} + \widehat{\beta}_{1,t}x_{1,x,t+1} + \cdots + \widehat{\beta}_{k,t}x_{q,x,t+1}\right)}
\tag{7}
$$

In that case, we have an under-dispersed model in the sense that, by construction, we must have $\text{Var}[Y_{z,t}] < \mathbb{E}[Y_{z,t}]$




- the negative binomial model, where $Y_{z,t} \sim NB(E_{z,t}, p_{z,t})$, where $E_{z,t}$ is the exposure and $p_{z,t}$ is the probability, with standard notations for the negative binomial probablity function. In that case, we have an over-dispersed model in the sense that $\mathrm{Var}[Y_{z,t}] > \mathbb{E}[Y_{z,t}]$

Calibrating the GLMs and averaging the indicators over the years spanning from 2001 to 2018, using the yearly cross-validation method, the results on the left of Table 2 were obtained. This table shows that the Negative Binomial model has the lowest AIC and BIC.

In order to better consider the characteristics of the claims data, Zero-Inflated models were tested using Poisson and Negative-Binomial distributions. In these models the joint use of logistic and count regression allows the integration of the
over-representation of non-events in the data. More formally, in Zero-Inflated models, given a location and a time $(z, t)$, we assume that there is a probably to have $0$ claims. In our context, it could mean that the town has not been recognized as hit by a drought event. The occurrence of a drought is modeled with a logistic model, with probability $p_{z,t}$, here. Then, if there is a drought event, the number of claims is driven by some specific distribution (Poisson, Binomial, or Negative Binomial), as introduced by Lambert (1992). If we consider some Poisson regression, it means that

$$\mathbb{P}(Y_{z,t} = 0) = p_{z,t} + (1 - p_{z,t})e^{-\lambda_{z,t} \cdot E_{z,t}} \tag{8}$$

(the second part comes from the fact there there could be a drought, but the number of counts of claims is null) and

$$\mathbb{P}(Y_{z,t} = y) = (1 - p_{z,t})\frac{[\lambda_{z,t} \cdot E_{z,t}]^y e^{-\lambda_{z,t} \cdot E_{z,t}}}{y!}, \text{ where } y = 1, 2, 3, \cdots \tag{9}$$

where $\lambda_{z,t}$ and $p_{z,t}$ are related to covariates through expressions as Equation (4) and (7), respectively. Note that such models can easily be estimated with standard statistical packages. The results of the Zero-Inflated models are visible on the right of
Table 2. It shows that the Zero-Inflated Negative-Binomial model is better than the Zero-Inflated Poisson model.

**Table 2.** Quality measures for the different GLM distributions.

| | Binomial | Poisson | NB | zero-inflated Poisson | NB |
|---|---|---|---|---|---|
| AIC | 115,051 | 114,189 | 100,491 | 71,154 | 54,375 |
| BIC | 115,113 | 114,252 | 100,564 | 71,259 | 54,510 |

Figure 3 shows the yearly predictions for the zero-inflated models, alongside the previously tested GLMs[3].

When looking at the yearly total predictions compared to reality, as observed in figure 3, all the GLM predictions are very similar apart from the negative binomial model which overpredicts (massively) in 2018, however they all overestimate claims
in 2018 and underestimating the number of claims in 2003, 2011, 2016 and 2017. This graph shows that the predictions closest

[3]For purposes of confidentiality, the total number of claims per year are withheld, but the proportions on the $y$-axis are valid.





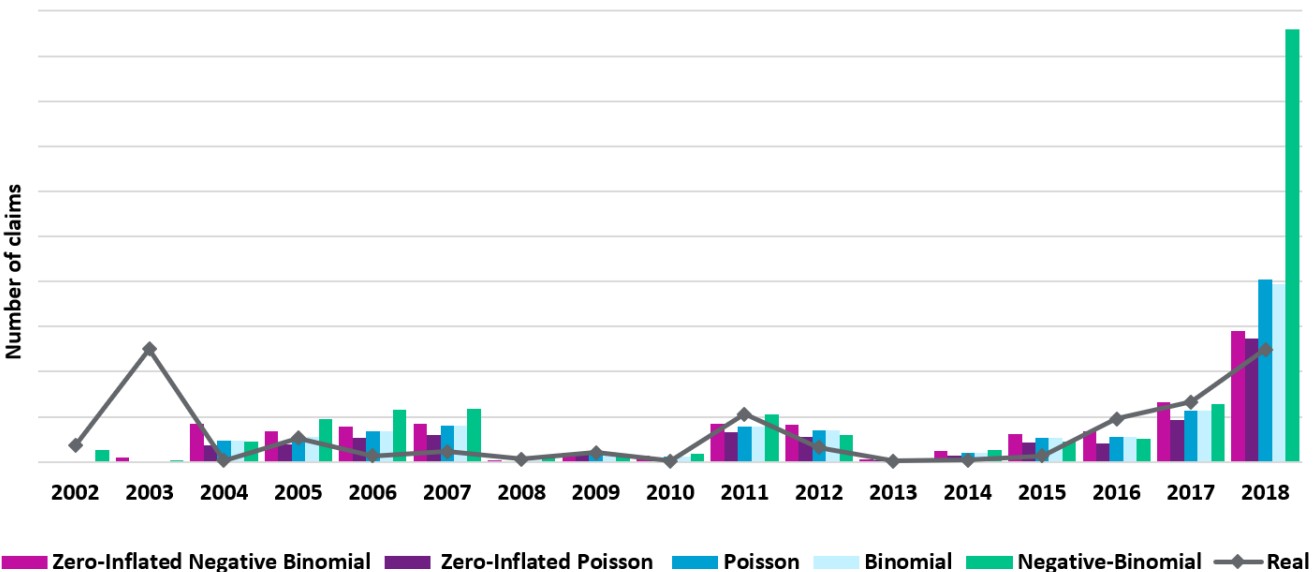

**Figure 3.** Yearly predictions for the Zero-Inflated models and GLMs

to the reality line are for the Negative-Binomial Zero-Inflated model. That model also has the best metrics when comparing to those of the GLMs. Thus, Zero-Inflated models appear to provide a better fit than the GLMs.

This section showed that the Zero-Inflated models, in particular the Negative-Binomial Zero-Inflated model, outperformed the GLMs in terms of number of claims and model selection criteria. In the next section, we will see of alternatives to

regression-type models can be considered, since the "*linear model*" assumption might be rather strong here.

### 3.4 Tree-based models

Tree-based models are popular models for data analysis and prediction and offer an alternative to the previous parametric models. Popularised by Breiman et al. (1983), regression trees produce simple and easily interpretable split rules.

– A Regression Tree is such that

$$
{}_t\widehat{Y}_{z,t+1} = \sum_{\ell=1}^{L} \widehat{\omega}_{\ell,t}\mathbf{1}(\boldsymbol{x}_{z,t+1} \in \mathcal{L}_\ell) \tag{10}
$$

where $\{\mathcal{L}_1, \cdots, \mathcal{L}_L\}$ is a partition of $\mathcal{X}$, and $\mathcal{L}_j$'s are called *leaves*. In a tree with two leaves, $\{\mathcal{L}_1, \mathcal{L}_2\}$, there is a variable $j$ such that $\mathcal{L}_1$ is the half-space of $\mathcal{X}$ characterized by $x_{j,t} \le s$ while $\mathcal{L}_2$ is characterized by $x_{j,t} > s$ for some threshold $s$. For the "classical" regression tree, the split is based on the squared loss function, in the sense that we select $s$ to maximise the between-variance, or equivalently, minimise the within-variance. It is possible to extend this approach by

using, instead of the squares of residuals (corresponding to the squared loss function), the opposite of the log-likelihood





of the data. This can be performed using the `rpart` R package (see Breiman et al. (1983) for further details on regression trees).

If trees are simple to interpret, they are usually rather unstable: when fitting a tree on subsets of observation, it is common to get different splitting variables, and therefore different trees. The idea of *bagging* (as defined in Breiman (1996)) is to use a
boostrap procedure to create samples (resampling the observations with replacement), and then to aggregate predictions. In the case where the number of covariates is not too large, this will be also called a Random Forests, from Breiman (2001).

Some Tree-Based models are tested in order to attempt to improve the previous predictions. Three tree-based approaches were used, here:

- A "classical" Random Forest (RF),

$$
{}_t\widehat{Y}_{z,t+1} = \frac{1}{m}\sum_{i=1}^{m} {}_t\widehat{Y}_{z,t+1}^{(i)} \text{ where } {}_t\widehat{Y}_{z,t+1}^{(i)} = \sum_{\ell=1}^{L_i} \widehat{\omega}_{\ell,t}^{(i)} \mathbf{1}(\boldsymbol{x}_{z,t+1} \in \mathcal{L}_\ell^{(i)}) \tag{11}
$$

where each tree – corresponding here to different $i$'s – is computed using a squared loss function, on different boostrap samples (obtained by resampling $n$ observations, with replacement, out of the initial $n$ ones). This can be performed using `randomForest` R package.

- A Poisson Random Forest (RFP) which considers count data, with the aim to better capture the distribution of the data.
Poisson Random Forests are a modified version of Breiman's Random Forest allowing the use of count data with different observation exposures. This is done by modifying the splitting criterion so that it maximised the decrease of the Poisson deviance and an offset has been introduced to accommodate for different exposures. This Random Forest was calibrated using the `rfPoisson` function available in the `rfCountData` package in R.

These Random Forests were all tuned in order to obtain optimal number of trees, number of variables tested at each split
and maximum number of nodes. However, the tuning was limited given the length of the tuning process, which may reduce the quality of the models.

The figure 4 shows the total yearly predictions for each tree-based model alongside the Zero-Inflated Negative Binomial model and the real claims.

Figure 4 shows that the closest predictions to the real observations (the plain line) seem to be those of the Zero Inflated
model and the Poisson Random Forest, although all models but the Zero-Inflated model underestimates 2011 and 2016. With our cross-validation approach, we have poor results for early years (only data from 2002 were used to derive a model for 2003). The standard Random Forest overpredicts 2018.Thus, the Zero-Inflated model and the Poisson Random Forest appear to have the best predictions.

Through this section, two different Random Forests were presented; however, the one with the best results is the Poisson
Random Forest which rendered similar results to the Negative-Binomial Zero-Inflated model. The complex calibration process and the unclear influence and importance of the variables on the output of the Poisson Random Forest make it a less attractive





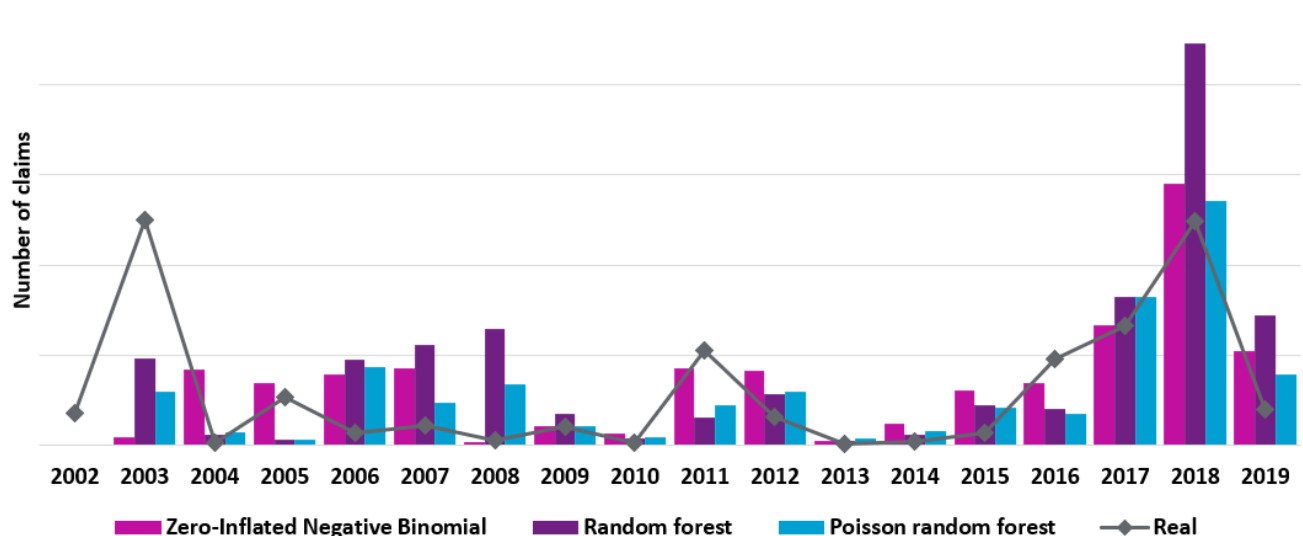

**Figure 4.** Comparison of yearly predictions for the tree-based models, with ZINB, classical RF, and RFP.

choice of model compared to the Zero-Inflated regression, which has a clear variable influence and a simple prediction formula. One can thus wonder whether such a loss in interpretability is worth such a small gain in term of predictions.

## 3.5 Mapping the predictions

In order to improve the predictions, and only predict the actually impacted areas, a methodology was developed to optimise the removal of all the very low claim predictions. This methodology was applied to both the Zero-Inflated model and the Poisson Random Forest. The total predictions changed little, and the geographical distribution of both models' predictions are observable for 2018 in figure 5.

   In 2018, both models have more or less the same claim distribution, with large amounts of claims along the Mediterranean, in
the North and in Pays-de-la-Loire. However, the distribution is slightly different when looking at the centre of France. Indeed, the Random Forest seems to predict more claims in that area than the Zero-Inflated Negative Binomial model. Comparing these results with the real claims for 2018, it can be seen that the historical claims are nearly exclusively concentrated in the centre of France with a few claims along the Atlantic and Mediterranean coast and in the North-East of France. Thus, both models seem to over predict the claims around the coasts of France and - slightly for the Poisson Random Forest and vastly for the
Zero-Inflated Negative Binomial model - under-predict in the centre of France. However, it can be noted that the over-predicted areas do mostly fall within areas that have non-recognised natural catastrophe declarations which could mean that the area was hit but not compensated and thus that the models have difficulty assessing the difference between areas that will be recognised as natural catastrophes, or not. The same conclusion can be made when looking at the predictions for 2017, visible in figure 6.





(a) Poisson RF

(b) Zero-Inflated

(c) Observed claims

(d) Nat.Cats.

**Figure 5.** Observed and Predicted number of claims for 2018



(a) Poisson Random Forrest (RFP)

(b) Zero-Inflated (ZINB)

(c) Observed claims

(d) Nat.Cats.

**Figure 6.** Observed and Predicted number of claims for 2017

In these maps, the predicted claims do appear to be in the correct areas in the South and South-West of France for both
models, however, there are overpredictions in the East, centre and North of France, which also appear to be areas with non
accepted natural catastrophe declarations. Both models seem to predict the correct areas but also additional zones that often





are areas that have refused natural catastrophe declaration, which means that those areas were impacted by subsidence but not sufficiently to enter into the scope of the natural catastrophe regime. Indeed, as the acceptance criteria changed frequently between 2001 and 2018, the models cannot capture the natural catastrophe aspect of a claim seeing as one that may have been acceptable in 2017 may no longer be today. Both models predict more or less the same geographical distribution of claims, posing the question of the usefulness and practicality of using a complex model such as the Poisson Random Forest compared to the Zero-Inflated Negative-Binomial model.

Another option that would permit the classification of the predicted claims into accepted and refuse natural catastrophe categories, would be to use the Shrinkage and Swelling of clay risk map Georisques (2020), published by Geological and mining research bureau (BRGM). This map categorises the exposure of a given point of France between four categories: not at risk, low risk, average risk and high risk. The acceptance of a natural catastrophe declaration in France is feasible if more than 3% of the surface of a town is in a zone with average or high risk. Thus, if the predicted claims map and the risk map, aggregated by town, were overlapped, the classification of the predicted claims would be possible between potentially accepted and most likely refused natural catastrophe declaration.

## 4 Cost predictions

In the previous section, we've seen different models to predict the frequency (the number of claims per town), and here, we consider some Gamma model for average cost per claim, leading us towards a *compound model* for the total cost per town, as introduced in Adelson (1966), and used for example in hydrology in Revfeim (1984) or Svensson et al. (2017), or for droughts in Khaliq et al. (2011).

### 4.1 Modeling average and total economic costs

For a given location $z$ and year $t$, the total cost (from the insurer's perspective) is a *compound sum*, in the sense that

$$
Y_{z,t} = \sum_{i=1}^{N_{z,t}} Z_{i,x,t} = \begin{cases} Z_{1,x,t} + \cdots + Z_{N_{z,t},x,t} \text{ if } N_{z,t} > 0 \\ 0 \text{ if } N_{z,t} = 0, \end{cases} \tag{12}
$$

with a random sum of random costs. Here $N_{z,t}$ is the frequency, modeled in the previous section, and $Z_{i,x,t}$'s are individual economic losses per house. It is possible to consider some Tweedie GLM, as introduced in Jørgensen (1997), corresponding to some compound Poisson model, with Gamma average cost. Nevertheless this is only a subclass of the general compound models. Note that Tweedie models are related to some *power-parameter*, since they are characterize by a relationship $\mathbb{E}[Y] = \mathrm{Var}[Y]^{\gamma}$. If $\gamma = 1$, $Y$ is proportional to some Poisson distribution (average costs are non-random) while $\gamma = 2$ means that $Y$ is proportional to some Gamma distribution (frequency is non-random). For inference, we used $\gamma = 1.5$ which corresponds the lowest AIC. For that Tweedie model, SSTI, SSWI, as well as Clay and Cat covariates were used.





If the model is interesting, it is less flexible than have two separate models, one for the frequency, and one for the average
cost at location $z$, for year $t$, and to write

$$Y_{z,t} = N_{z,t} \cdot \overline{Z}_{x,t} \tag{13}$$

where (as before) $N_{z,t}$ is the frequency, as modeled in the previous section, and we can use our data, aggregated at the town
level, to model the average cost (per house) $\overline{Z}_{x,t}$. The prediction is then

$${}_t\widehat{Y}_{z,t+1} = {}_t\widehat{N}_{z,t+1} \cdot {}_t\widehat{Z}_{z,t+1} \tag{14}$$

The results of these methods are rather similar. Figure 7 shows the yearly predictions for the Tweedie model and the average
cost of claims method using the GLM and both the Zero-Inflated and Poisson Random Forest models, previously calibrated.

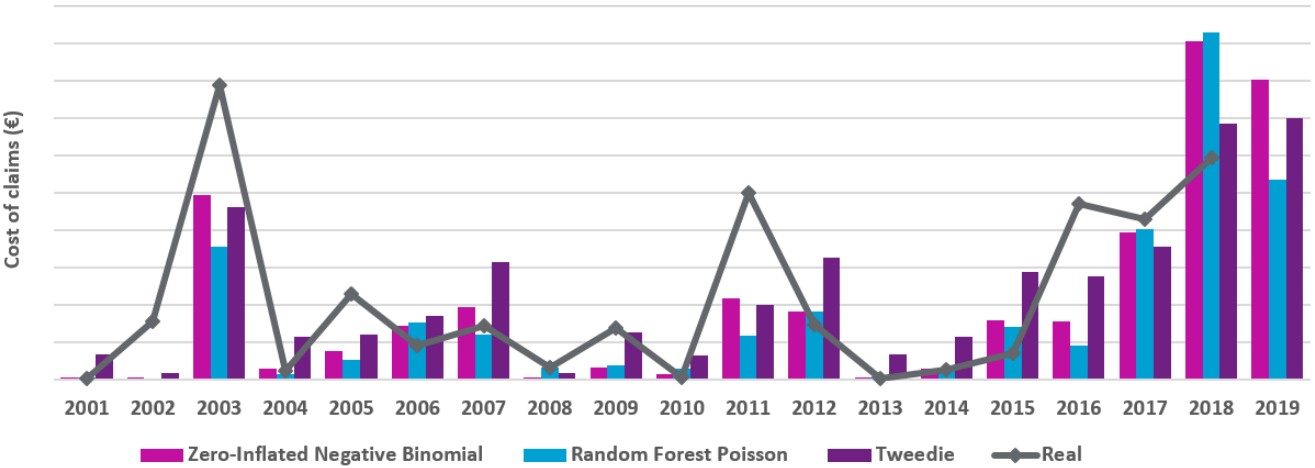

**Figure 7.** Yearly cost predictions, with a ZINB model + Gamma costs, RFP model + Gamma costs, and a Tweedie model (Poisson + Gamma
costs).

Figure 7 shows that the Tweedie model makes predictions that are similar to the previous cost of claims predictions, however
this model over-estimates less the year 2018 and (clearly) over-estimates more the years 2007 and 2012. On the other hand, the
years 2003, 2011 and 2016 are still severely underestimated by the three predictions, although less so by the Tweedie model.

## 4.2    Mapping the predictions

As in section 3.5, it is possible to visualize the prediction, and to map ${}_{2016}\widehat{Y}_{z,2017}$ and ${}_{2017}\widehat{Y}_{z,2018}$, on Figure 8. As expected, if
we are not able to predict correctly the frequency, the cost is overestimated. Overall (as we can see on Figure 7), in 2017, we
obtained a good prediction, in France, but we can visualize some spatial differences. Most of the comments made in section
3.5 remain valid, and clearly, predicted the economics losses in a specific area is not a simple task.





(a) Real 2017

(b) Predicted 2017

(c) Real 2018

(d) Predicted 2018

**Figure 8.** Observed and Predicted average cost of claims in 2017 and 2018



## 5    Conclusions

The increase in number and severity of subsidence claims in the past years has created a need for insurers to better grasp their knowledge of this risk. However, the implementation of subsidence models is time consuming and requires detailed data. This study proposed a method of approaching the costs and frequency of claims due to subsidence based on historical data.

This was applied through two main components: the development of new drought indicators using Open Data and the use of parametric and tree-based models to model this risk. Modelling subsidence requires the integration of meteorological and geological indicators to ascertain the factors predisposing a policy to subsidence. Without this information, the inherent risk to which a dwelling is exposed cannot be perceived. For this reason, geological and meteorological data was obtained from Open Data sets. This data was paired with insurance exposure and claim data to obtain a complete dataset used to calibrate the

models. However, the data available was only at a communal mesh, making the results less precise. Indeed, subsidence is a very localised risk whose modelling would benefit from policy level data.

Overall, the methods enable the predictions of an estimate of the number of claims, cost of claims and their geographical distributions. Although they sometimes lack in precision, the models give a good indication of the severity of a given year. The uncertainty in the predictions may be explained by the non-homogeneous data on which the models were calibrated. Indeed,

the natural catastrophe declaration criteria evolved many times over the calibration period, rendering the historical data biased. The predictions would also have benefited from data at a finer mesh to take into account the individual particularities of each policy such as the presence of trees close to the construction, the slope of the terrain, the orientation of the slope, etc. Finally, more precise temporal definition of the events could have been used if the claims were available at a monthly scale rather than yearly. In that case, seasonal indicators could have been created, rather than yearly ones, to capture more precisely the time

scale of an event.

This study has allowed for the creation and implementation of new drought indicators for a subsidence claim model which performs well considering the limitations induced by the lack of information and precision of the calibration data set. The created model provides a better understanding of the phenomenon and allows other subsidence exposure evaluations to be challenged which will improve an insurance company's management of this risk, given the lack of available subsidence models

on the market. This methodology could be improved in the future by using policy level data with a more accurate temporal definition of the claims in order to better apply the created indicators. The framework of this study is also in itself interesting and could be repurposed for other applications. Indeed, the studied models could be extended to other claim prediction problematics, which are often confronted with similar issues of over representation of non-recognized events.

*Author contributions.*  HA provided insurance data and provided technical support. MJ developed the model code and performed the simulations. AC prepared the manuscript with contributions from all co-authors





*Competing interests.* The authors declare that they have no conflict of interest

*Acknowledgements.* The authors wish to thank Hélène Gibello, Sonia Guelou, Florence Picard and Franck Vermet for comments on a previous version of the work. Arthur Charpentier received financial support from the Natural Sciences and Engineering Research Council of
Canada (NSERC-2019-07077) and the AXA Research Fund.



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
