# Peer review of "Predicting Drought and Subsidence Risks in France"

_Natural Hazards and Earth System Sciences, 2021_

## Author Response (AR1)

*Response to RC1*

dear reviewer, thanks a lot for your comments. I did not find the zip folder, so I could not provide further comments. All apologies

*Response to RC2*

1) "**The article should clearly discuss how drought conditions increase groundwater demand and how groundwater withdrawal is affects clay layer causing subsidence**"
Those are indeed important in the US (https://www.usgs.gov/special-topics/water-science-school/science/land-subsidence) or in Asia, such as in Vietnam (https://hal.archives-ouvertes.fr/hal-01888487/document) or Jakarta (https://doi.org/10.1016/j.ocecoaman.2021.105775), but such a phenonema is not reported in France.

2) "**Evapotranspiration (ET) is an important component of the hydrologic cycle which has not been incorporated in these variables**"
Indeed, our variables do not take into account ET *explicitly* (we only consider soil moisture, soil temperature and precipitation), even if a correlation undoubtedly exists between our variables and ET because we consider soil heat, moisture and precipitation.

Other interesting indicators can indeed be constructed by adding ET (e.g. SPEI, more powerful than SPI), however this indicator is sensitive to the method of calculation of "potential evapotranspiration". And there were granularity issues with the data, that were not on the same scale as other variables. This is why we did not incorporate that component explicitely.

3) "**no error metrices have been presented to represent each model performance compared to original observations**"
Here a a summary of various statistics,

|  | TPR (%) | Gini (%) | RMSE (%) | AIC | BIC |
|---|---|---|---|---|---|
| Binomial | 18.5% | 84.0% | 0.0080 | 115,051 | 115,113 |
| Poisson | 18.5% | 92.7% | 0.0081 | 114,189 | 114,252 |
| Quasi-poisson | 17.8% | 92.7% | 0.0081 |  |  |
| Negative-Binomial | 20.9% | 94.1% | 0.0142 | 100,491 | 100,564 |
| ZI Poisson | 14.9% | 93.2% | 0.0079 | 71,154 | 71,259 |
| ZI Negative-Binomial | 18.1% | 93.4% | 0.0080 | 54,375 | 54,510 |
| RFF | 16.7% | 91.5% | 0.0083 |  |  |
| RF poisson | 15.2% | 91.5% | 0.0079 |  |  |

Such a table was not incorporated since we are not really confortable with those measures on *counting* variables (related to some Poisson loss). There are standard measures, with pros and cons, for continuous variables (RMSE) or binary ones (TPR, Gini), but no real consensus on counting variables. We can add that table if necessary

---

## Referee Report (RR1)

**General comments**

- The content of the article is very novel and addresses a concrete societal problem: the difficulties in predicting building damage from subsidence, an increasingly costly climate risk for insurers. In some countries, such as France, this risk is insured via household policies. The methodology and results will be useful for the insurance industry in France, and may also be useful in other countries with similar established insurance products, and countries where insurance industry and policy makers investigate the possibly for developing insurance products for this risk. Overall the paper is well-structured. It compares various statistical models using a range of indicators to predict subsidence claims, calibrated against a historical dataset from insurance companies.

- I understand a forecasting exercise may not fit within the scope of this article, but I feel the article would benefit from a short discussion on what would be needed to use the models to forecast damage claims in the future, e.g. in terms of data and under different climate and economic development scenarios etc.

- At the risk of shamefully promoting my own work, I do think this conference paper https://piahs.copernicus.org/articles/382/577/2020/ is highly relevant to the context of your study – we developed a predictive risk model for subsidence and groundwater-related damage in the Netherlands (sadly lacking any temporal component related to drought). It is not statistical in the absence of an insurance industry for subsidence risk and consequent lack of any data regarding historical damage. I think particularly your paper would benefit from a bit more explanation on the exposure/ vulnerability of buildings as the focus is now heavily on the hazard side of the risk: this aspect features more strongly in our approach.

- Related to grammar and spelling:
  - Perhaps a matter of style, but  I feel this article overall suffers a bit from overuse of commas.
  - When referring to an article by multiple authors (e.g.  Iglesias et al.), the subject of the sentence is plural, which should be reflected in the verb: Iglesias et al. (2019) *point out*, rather than *points out*. Adjust throughout article.
  - Please have a good read-through to check for spelling – particularly regarding plural/ singular forms there are still quite a few mistakes throughout.

**Abstract**

- Line 7: 'showing that climate change will probably have.. on this risk': perhaps I missed it in the paper, but do you indeed show the impact of climate change on the development of the subsidence risk using the statistical models? What climate scenarios are you using?

**1. Introduction**

- Section 1.1 title: 'Drought and climate change': this paragraph also covers economic impact of drought and availability of insurance products against droughts. Perhaps change title of paragraph to reflect this?
- Line 29-30: what is the point you want to make by mentioning the publication of Hagenlocher et al? What are the results of this literature review?
- Line 30-34: what kind of drought impacts are studied by Naumann et al? Does their estimate also relate to subsidence/ buildings or mostly other drought impacts (e.g. navigation, agriculture, drinking water)?
- Line 34: what do you mean by 'still two times larger when expressed to the relative size of the economy.'?  Gross or net national product?

- Line 30-34: check grammar of sentence, e.g. change 'they provide some forecasts…' to . 'they forecast that, under the absence of climate action, annual drought losses…. '
- Line 35-36: what do you mean by a 'socio-economic factor' persisting and accelerating? Do you mean perhaps that current drought trends and corresponding economic risks will further accelerate (or perhaps better: aggravate?) under climate change?
- Line 43: '..insurance products which does not' should be do
- Line 45: 'in most country > countries ; .. 'as such as frost': what do you mean by this? Do you mean drought is part of a package of weather-related events against which you can insure crops (including frost)?
- Line 49: 'we will use data from..': is discussed in chapter 2, suggest to keep to explaining impact of droughts on subsidence in this para. Also, the risk you are quantifying is not clay shrinkage-induced subsidence (that is the hazard!), but the risk of building damage due to clay shrinkage subsidence.  Risk = likelihood of event (e.g. drought/ severity of subsidence) * exposure (buildings) * impact (degree of damage to building/ restoration costs). Without buildings to suffer from the subsidence, there is no risk of damage.  In turn, the risk of building damage is only part of the wider scope of economic impacts of this (drought-induced) subsidence (as described in Kok and Costa 2021)
- Line 75-76: ''Indeed… in place': this is a bit hard to follow. Can you change to just say 'since subsidence coverage has been in place, xx % of claims has been related to subsidence?
- 80-102. Perhaps a short description of the physical process of damage and restoration (I assume that's what the claims are made for) would be valuable in this section, as well as a discussion (perhaps in section discussion) about whether it is reasonable to assume all buildings in at-risk zones will be damaged and restored once in the next decennia, or multiple times. Are all buildings equally susceptible or are there differences (e.g.  newer buildings better prepared against subsidence risk)? And if that is the case, would this mean that over time the risk (due to less/ no damage resulting from subsidence event) will also decrease (even if the hazard increases)?
- Section 1.3: Aside from the generic study purpose and set up, would expect to also read something about how the work relates to the knowledge gap (have such models been used before and/or in a similar context?) as well as the potential application/ value of the models by the insurance market (how does it relate to current practice?) and possibly others (e.g. to inform infrastructure asset managers of subsidence risk). Could also be addressed in discussion/ conclusion.

**Chapter 2**

- 126: what do you mean by 'inherent risks of the town' ?
- Line 131: are the three companies' insurance policies equally spread throughout the country/ at-risk areas? i.e. do they present a geographically representative sample of the whole market? Perhaps add something on this
- Line 168-170; Are there any thresholds identified for when a drought or subsidence event is deemed abnormal? E.g. 1: 300 year return event? And also, I imagine that with droughts predicted to occur more often under climate change (as indeed already seen in the past decades), it is reasonable to expect a further decline in proportion of acceptance of natural catastrophe recognition requests?  (Note: I see this is addressed in section 2.2. Perhaps refer to this section or merge these sections?)
- Line 175: By 'order' you mean request for recognition as natural catastrophe?

- Line 196: 'humidity' is usually used in context of moisture content in the air. For soil it is more common to use 'soil moisture content', when referring to the unsaturated zone above groundwater table. Is this what is meant? And where does the groundwater table factor in (e.g. groundwater with high variability poses higher risk than low variability).
- Line 199: based on 'the indicator is calculated for every months based on the indicators of the three previous months', shouldn't July be determined using the mean of April, May and June?
- Line 208-216: what do all these updates mean for the likeliness of a (similar) event being recognized as natural catastrophe?
- Line 245: ''sole use of soil moisture' > which indicator are you referring to now? I thought SPI (in section above) uses only precipitation data?
- 246-249: So in the case of 2003, SSWI shows a drought, but SPI does not? This sentence is not very easy to understand.
- 286: 'but they were less predictive than the maximum': can you give any explanation as to why? It seems using the highest concentration would lead to an overestimate of claims, but apparently not ?

**Chapter 3**

- Line 325: bullet points: what are you listing here? Suggestions for others on what to use in case of spatio-temporal data, or is this what you will be doing? Clarify in writing.
- Line 356-357: 'if we predict… of each variable.' This is difficult to follow. Can you elaborate and/ or provide an example here?
- Line 410: 'three tree-based approaches were used': but you only list two.

**Chapter 4**

- 470-481: I think when you write 'some' you just mean 'a'?
- 475-479: 'if the model is interesting… average cost' this sentence is hard to follow.
- Figure 7: Is it possible to fit a scale to the Y-axis? And are these costs total claims per year in France, costs of claims per town or per individual claim?

**5. Conclusions**

- 500-501 ' ..grasp their knowledge of this risk' not sure if this is is good english. Perhaps change to 'improve their grasp of the knowledge of this risk'
- 502: In this paper, you propose a method for predicting claims. Do these insurers not use any claim prediction models at all at the moment? What does current practice look like for them?
- 509: 'policy level data': what is this? From my experience, risk data at the building level (though it would be highly valuable) is also not (freely) available in policy context due to its sensitive nature.
- 525: can you give examples of similar claim prediction difficulties for which the models discussed in this study could be used?

---

## Author Response (AR2)

**Response to the Editors of *NHESS* and to the reviewers**

June 2, 2022

Ref. : "*nhess-2021-214*"

Dear Editors and reviewers,

Please find enclosed a new revised version of our article. Following suggestions you have made on the previous version, we made substantial changes in the manuscript, major changes within (new paragraphs) are highlighted in red. Minor changes are not highlighted in the manuscript.

As you will see in this letter, we have taken into account the many suggestions and remarks you made. We hope that the changes we have made will convince you of the value of this work. For ease of reading, in this letter, we have adopted the following visual codes:

black texts: comments from the reviewers

text in blue frames: our answers to those comments and questions

**RC1**

**General comments**

The content of the article is very novel and addresses a concrete societal problem: the difficulties in predicting building damage from subsidence, an increasingly costly climate risk for insurers. In some countries, such as France, this risk is insured via household policies. The methodology and results will be useful for the insurance industry in France, and may also be useful in other countries with similar established insurance products, and countries where insurance industry and policy makers investigate the possibly for developing insurance products for this risk. Overall the paper is well-structured. It compares various statistical models using a range of indicators to predict subsidence claims, calibrated against a historical dataset from insurance companies.

Thanks a lot.

I understand a forecasting exercise may not fit within the scope of this article, but I feel the article would benefit from a short discussion on what would be needed to use the models to forecast damage claims in the future, e.g. in terms of data and under different climate and economic development scenarios etc.

That is a great idea, unfortunately, it would be difficult for several reasons. Since it is something readers might expect, we added the following paragraph at the end of section 4.2 (before the conclusion) *in order to forecast claims in the future, we would have to get forecasts of future climate dataset for all three climatic indicators (ESSTI, ESSWI and ESPI) using the soil wetness, soil temperature and precipitation data in different RCP scenarios and use these new future climate indicators to estimate the impact of climate change on future claims in each scenario. Furthermore, one must keep in mind that the results would be base on todays' legal environment and are barring any evolution in the way subsidence claims are covered. This methodology would also be flawed by the generally larger granularity of future climate data, which renders the estimation less precise (eg. CMIP5 future climate data (https://cds.climate.copernicus.eu/cdsapp#!/dataset/projections-cmip5-monthly-single-levels?tab=overview) which has lower resolution compared to ERA5 land data used in this study).*

At the risk of shamefully promoting my own work, I do think this conference paper – Costa et al. (2020) – is highly relevant to the context of your study – we developed a predictive risk model for subsidence and groundwater-related damage in the Netherlands (sadly lacking any temporal component related to drought). It is not statistical in the absence of an insurance industry for subsidence risk and consequent lack of any data regarding historical damage. I think particularly your paper would benefit from a bit more explanation on the exposure/ vulnerability of buildings as the focus is now heavily on the hazard side of the risk: this aspect features more strongly in our approach.

Thanks for pointing out the reference, that we missed. We did remove some lines (64 - 69) and added: *Subsidence risk is defined (Ministère de la transition écologique et solidaire (MTES) (2016)) as the displacement of the ground surface due to shrinkage and swelling of clayey soils. It can cause damage to buildings and infrastructures as it creates misalignments in the foundations of buildings if the soil shrinks at a different rate all over. Due to their superficial foundations, individual houses are particularly prone to this risk. Although largely caused by long term meteorological events, subsidence is also due other important predisposing factors such as the type of soil. Clayey soils have the particularity of having a very different consistency depending on its moisture: Hard and crackly when dry and very malleable when wet. These modifications are twinned with a variation in volume: very high volumes when wet (swelling) and low volumes when dry (shrinkage). Therefore, the depth, width and concentration of clayey soils plays an important role in the realisation of this phenomenon. France, having a temperate climate, has saturated clayey soils, making subsidence predominant during droughts. However,*

*other factors can also contribute to this risk. Indeed, the quality of the construction will have an important impact on its exposure. As mentioned previously, constructions with shallow foundation are particularly prone to this risk. The impact and type of degradation caused by subsidence also varies with the age of the construction, the absence or presence of a basement, the types of material used for the foundations and their depth. Indeed Costa et al. (2020) show that, in the Netherlands, the vulnerability of buildings and the severity of subsidence can be characterised by those variables.*

Related to grammar and spelling:

- Perhaps a matter of style, but I feel this article overall suffers a bit from overuse of commas.

- When referring to an article by multiple authors (e.g. Iglesias et al. (2019)), the subject of the sentence is plural, which should be reflected in the verb: Iglesias et al. (2019) point out, rather than points out. Adjust throughout article.

- Please have a good read-through to check for spelling – particularly regarding plural/ singular forms there are still quite a few mistakes throughout.

All apologies. We did remove some commas, and did check for spelling and for plural / sigular issue with references.

**Abstract**

Line 7: 'showing that climate change will probably have.. on this risk': perhaps I missed it in the paper, but do you indeed show the impact of climate change on the development of the subsidence risk using the statistical models? What climate scenarios are you using?

The end of the sentence has been removed.

**1. Introduction**

Section 1.1 title: 'Drought and climate change': this paragraph also covers economic impact of drought and availability of insurance products against droughts. Perhaps change title of paragraph to reflect this?

Yes, thanks, it was modified to : *Droughts, economic impacts, insurance coverage and climate change*.

Line 29-30: what is the point you want to make by mentioning the publication of Hagenlocher et al. (2019)? What are the results of this literature review?

We added a sentence in the revised version of the paper: *despite major advances over the past decades in terms of developing better methods and tools for characterizing individual components of risk, Hagenlocher et al. (2019) mentioned persistent knowledge gaps which need to be confronted in order to advance the understanding of drought risk for people and policymakers, to move towards a more drought resilient society.*

Line 30-34: what kind of drought impacts are studied by Naumann et al. (2021)? Does their estimate also relate to subsidence/ buildings or mostly other drought impacts (e.g. navigation, agriculture, drinking water)?

This relates to other drought impacts such as agricultural losses. We did add a sort note in the revised version of the paper : *drought damages, not related to subsidence or building damages but mainly agricultural losses, could strongly increase with global warming and cause a regional imbalance in future drought impacts*

Line 34: what do you mean by 'still two times larger when expressed to the relative size of the economy.'? Gross or net national product?

Sorry, it is based on the GDP. Actually that part was a direct citation from Naumann et al. (2021) (information related to "relative size of the economy" as % of GPD are intensively discussed). We wrote in the revised version *still "two times larger when expressed to the relative size of the economy" (expressed as a fraction of the GDP).*

Line 30-34: check grammar of sentence, e.g. change 'they provide some forecasts...' to . 'they forecast that, under the absence of climate action, annual drought losses.... '

Sorry, change made.

Line 35-36: what do you mean by a 'socio-economic factor' persisting and accelerating? Do you mean perhaps that current drought trends and corresponding economic risks will further accelerate (or perhaps better: aggravate?) under climate change?

> Yes, that is what Bevere and Weigel (2021) suggests. In the revised version, we wrote *Bevere and Weigel (2021) suggests that current drought trends and corresponding economic risks will further aggravate under climate change.*

Line 43: '..insurance products which does not' should be do

> Again, all apologies for the typos. Change made.

Line 45: 'in most country' > countries ; .. 'as such as frost': what do you mean by this? Do you mean drought is part of a package of weather-related events against which you can insure crops (including frost)?

> A short sentence has been added in the revised version *For example in Spain, it is possible to insure rain-fed crops against drought, as discussed in Entidad Estatal de Seguros Agrarios (ENESA) (2012), but in most countries, drought coverage only concerns agricultural (crop) insurance, in the same way that frost is covered.*

Line 49: 'we will use data from..': is discussed in chapter 2, suggest to keep to explaining impact of droughts on subsidence in this para. Also, the risk you are quantifying is not clay shrinkage-induced subsidence (that is the hazard!), but the risk of building damage due to clay shrinkage subsidence. Risk = likelihood of event (e.g. drought/ severity of subsidence) * exposure (buildings) * impact (degree of damage to building/ restoration costs). Without buildings to suffer from the subsidence, there is no risk of damage. In turn, the risk of building damage is only part of the wider scope of economic impacts of this (drought-induced) subsidence (as described in Kok and Costa (2021))

> Thanks, we replaced risk with peril. *In this article, we will use data from several insurance companies in France, regarding a very specific drought related peril, that is clay shrinkage induced subsidence which causes damage to buildings. Without buildings to suffer from subsidence, there is no risk of damage. In turn, the risk of building damage is only part of the wider scope of economic impacts of this (drought-induced) subsidence (as describe in Kok and Costa (2021)).*

Line 75-76: ''Indeed... in place': this is a bit hard to follow. Can you change to just say 'since subsidence coverage has been in place, xx % of claims has been related to subsidence?

Yes, we can, *Indeed, 37% of the total costs of natural catastrophes in France between 1982 and 2020 are caused by subsidence, 38% of which is concentrated over the period 2015-2019, which is 15% of the total time that subsidence coverage has been in place (as discussed in Mission des Risques Naturels (MRN) (2021)).*

80-102. Perhaps a short description of the physical process of damage and restoration (I assume that's what the claims are made for) would be valuable in this section, as well as a discussion (perhaps in section discussion) about whether it is reasonable to assume all buildings in at-risk zones will be damaged and restored once in the next decennia, or multiple times. Are all buildings equally susceptible or are there differences (e.g. newer buildings better prepared against subsidence risk)? And if that is the case, would this mean that over time the risk (due to less/ no damage resulting from subsidence event) will also decrease (even if the hazard increases)?

That is a very good point. *In 2007, plans (called PPR) for the Prevention of Differential Settlement Risks were prescribed in more than 1500 communes, as mentioned in Ministère de la transition écologique et solidaire (MTES) (2016). These plans are addressed in particular to anyone applying for a building permit, but also to owners of existing buildings. Its objective is to delimit the zones exposed to the phenomenon, and in these zones, to regulate the occupation of the land. It thus defines, for future construction projects and, if necessary, for existing buildings (with certain limits), the mandatory or recommended constructive rules (but also related to the environment near the building) aimed at reducing the risk of disorders appearing. In exposed sectors, the plan may also require a specific geotechnical study to be carried out, in particular prior to any new project. For the time being, therefore, these plans do not provide for dinconstructibility. Among the advice given to minimize the risk of doccurrence and lample of the phenomenon, there are instructions relating to the realization of a waterproof belt around the building, to the distance of vegetation from the building, to create a root barrier, to connect the water networks to the collective network, to seal the buried pipes, to limit the consequences of a heat source in the basement or to create a drainage device. There is also advice on how to adapt the building so as to counter the phenomenon and thus minimize the damage as much as possible, essentially by adapting the foundations (adopting a sufficient depth of anchorage, adapting according to the sensitivity of the site to the phenomenon, avoiding any dissymmetry in the depth of anchorage and preferring continuous and reinforced foundations, concreted to the full height of the excavation), by making the building structure more rigid (requiring the implementation of horizontal (top and bottom) (requiring the implementation of horizontal (top and bottom) and vertical (corner posts) ties for the connected load-bearing walls) or the disassociation of the various structural elements (by the installation of a rupture joint (elastomer) over the entire height of the building (including the foundations). building (including the foundations)).*

Section 1.3: Aside from the generic study purpose and set up, would expect to also read something about how the work relates to the knowledge gap (have such models been used before and/or in a similar context?) as well as the potential application/ value of the models by the insurance market (how does it relate to current practice?) and possibly others (e.g. to inform infrastructure asset managers of subsidence risk). Could also be addressed in discussion/ conclusion.

That is a fair question. To our knowledge, there have been either studies extremely located (geographically), such as Doucet (2018), or Corti et al. (2009), published more than 10 years ago. There is clearly a knowledge gap, but it is rather difficult to address that issue. That might be a topic for a dedicated paper, probably. A brief sentence was added in the conclusion: *xxxxxxxxxxxxxx*

**Chapter 2**

126: what do you mean by 'inherent risks of the town' ?

Sorry, we used 'inherent risk' to describe geophysical exposure. We removed, in the revised version 'inherent risk' and replaced it with : *creating the need for additional information about the climatic and geophysical exposure of a given town*.

Line 131: are the three companies' insurance policies equally spread throughout the country/ at-risk areas? i.e. do they present a geographically representative sample of the whole market? Perhaps add something on this

Yes, we added a short note *(those last two based on data from three important insurance companies in France, with exposures uniformly spread throughout the country, that could be seen as a geographically representative sample, representing about 20% of the French market)*.

Line 168-170; Are there any thresholds identified for when a drought or subsidence event is deemed abnormal? E.g. 1: 300 year return event? And also, I imagine that with droughts predicted to occur more often under climate change (as indeed already seen in the past decades), it is reasonable to expect a further decline in proportion of acceptance of natural catastrophe recognition requests? (Note: I see this is addressed in section 2.2. Perhaps refer to this section or merge these sections?)

Again, a short note as been added: *because the subsidence claim system is based on a 25 year return period threshold (as detailed in section ??), communes with recurring subsidence events would no longer be considered abnormal and their request will be declined.*

Line 175: By 'order' you mean request for recognition as natural catastrophe?

Sorry for the misunderstanding, we replaced 'order' with 'request'.

Line 196: 'humidity' is usually used in context of moisture content in the air. For soil it is more common to use 'soil moisture content', when referring to the unsaturated zone above groundwater table. Is this what is meant? And where does the groundwater table factor in (e.g. groundwater with high variability poses higher risk than low variability).

> Sorry, we replaced 'humidity' with 'moisture'.

Line 199: based on 'the indicator is calculated for every months based on the indicators of the three previous months', shouldn't July be determined using the mean of April, May and June?

> We changed the previous sentence *moisture indicator is calculated for every month based on the average of the indicator of that month and the two previous months.*

Line 208-216: what do all these updates mean for the likeliness of a (similar) event being recognized as natural catastrophe?

> Since the return period is defined locally, and not nationally, of the kind of "(similar) event" we talk about. Locally, a "(similar) event" will be less likely to be recognized as a "natural catastrophe". As mentioned in Article L 125-1 paragraph 3 of the Insurance Code (July 13, 1982), discussed in section 2.1, are considered as the effects of natural disasters "uninsurable direct material damage caused by the abnormal intensity of a natural agent, when the usual measures to be taken to prevent such damage could not prevent their occurrence or could not be taken" . Thus, it has always been claimed, if an event is similar to one that occurred the year before, it would be less likely seen as having had as a determining cause the "abnormal intensity" of a natural agent. This is why we wrote *The presence of a threshold set at a 25-year return period indicates that over time, if a commune is regularly hit by extreme droughts, those events will be less and less likely to be declared natural catastrophes as they will lose their exceptional character.*

Line 245: 'sole use of soil moisture' : which indicator are you referring to now? I thought SPI (in section above) uses only precipitation data?

> We changed to *similarly, the sole use of soil moisture, makes the SSWI a simple indicator to use, but also a deficient one*.

246-249: So in the case of 2003, SSWI shows a drought, but SPI does not? This sentence is not very easy to understand.

sorry. We changed it as follows: *for example they show that the droughts of 2003 are not linked to an extreme precipitation deficit. Those droughts could not be detected through the SPI, but could be using the SSWI.*

286: *'but they were less predictive than the maximum'*: can you give any explanation as to why? It seems using the highest concentration would lead to an overestimate of claims, but apparently not ?

About the "can you give any explanation as to why?", no: we simply used different statistics as explanatory features, and the maximum provided "better" prediction, based on the mean squared prediction error. I.e. "on average" it provides better predictions using the maximum, even if prediction might be more variables. We added "*(based on the mean squared prediction error)* in the revised version of the paper. Unfortunately, we have no clear explanation as to why.

**Chapter 3**

Line 325: bullet points: what are you listing here? Suggestions for others on what to use in case of spatio-temporal data, or is this what you will be doing? Clarify in writing.

In order to clarify, we added *In the case of spatio-temporal data, two approaches can be used,* and at the end of the second approach, we explicitly stated: *this approach is used in this study*.

Line 356-357: *'if we predict... of each variable'*. This is difficult to follow. Can you elaborate and/ or provide an example here?

All apologies, we changed for *if we predict more claims due to subsidence in time, it is mainly due to the underlying factors than to the change in the impact of each variable.*

Line 410: *'three tree-based approaches were used'*: but you only list two.

Fair point, indeed, it was 'two' tree-based approaches. Sorry.

**Chapter 4**

470-481: I think when you write 'some' you just mean 'a'?

> Yes, indeed, it was changed for 'a'.

475-479: *'if the model is interesting… average cost'* this sentence is hard to follow.

> Sorry, it was changed for *Although the model is interesting, it is less flexible than having two separate models, one for the frequency, and one for the average cost at location z, for year t, written :* (etc).

Figure 7: Is it possible to fit a scale to the *Y*-axis? And are these costs total claims per year in France, costs of claims per town or per individual claim?

**5. Conclusions**

> We changed the description for more details *Total yearly cost predictions in France*. Unfortunately, for privacy issue, no scale was possible as the data has to remain completely anonymous for our study.

500-501 ' ..grasp their knowledge of this risk' not sure if this is is good english. Perhaps change to 'improve their grasp of the knowledge of this risk'

> Yes, indeed, it was changed in the revised version to: *the increase in number and severity of subsidence claims in the past years has created a need for insurers to improve their grasp of the knowledge of this risk.*

502: In this paper, you propose a method for predicting claims. Do these insurers not use any claim prediction models at all at the moment? What does current practice look like for them?

> There is little model coverage for subsidence as it is covered by the French Natural Catastrophe Regime.

509: 'policy level data': what is this? From my experience, risk data at the building level (though it would be highly valuable) is also not (freely) available in policy context due to its sensitive nature.

> Indeed, those are not available freely no, an insurer could reproduce this study using their own policy level data though.

525: can you give examples of similar claim prediction difficulties for which the models discussed in this study could be used?

> Yes, we added *Indeed, the studied models could be extended to other claim prediction problematics, for other types of perils for example, which are often confronted with similar issues of over representation of non-recognized events.*
>
> Many thanks for all the comments and feedback !

**References**

Bevere, L. and Weigel, A. (2021). Exploring the secondary perils universe. *Swiss Re Sigma*, (1/21).

Corti, T., Muccione, V., Köllner-Heck, P., Bresch, D., and Seneviratne, S. I. (2009). Simulating past droughts and associated building damages in france. *Hydrology and Earth System Sciences*, 13(9):1739–1747.

Costa, A. L., Kok, S., and Korff, M. (2020). Systematic assessment of damage to buildings due to groundwater lowering-induced subsidence: Methodology for large scale application in the netherlands. *Proceedings of the International Association of Hydrological Sciences*, 382:577–582.

Doucet, S. (2018). *Combinaison de mesures géodésiques pour l'étude de la subsidence: application à la saline de Vauvert, Gard, France*. PhD thesis, Montpellier.

Entidad Estatal de Seguros Agrarios (ENESA) (2012). La sequía, un riesgo incluido en los seguros agrarios. *Noticias Del Seguro*, (82):3–5.

Hagenlocher, M., Meza, I., Anderson, C. C., Min, A., Renaud, F. G., Walz, Y., Siebert, S., and Sebesvari, Z. (2019). Drought vulnerability and risk assessments: state of the art, persistent gaps, and research agenda. *Environmental Research Letters*, 14(8):083002.

Iglesias, A., Assimacopoulos, D., and van Lanen, H., editors (2019). *Drought: Science And Policy*. Wiley-Blackwell.

Kok, S. and Costa, A. (2021). Framework for economic cost assessment of land subsidence. *Natural Hazards*, 106(3):1931–1949.

Ministère de la transition écologique et solidaire (MTES) (2016). *Le retrait-gonflement des argiles: Comment prévenir les désordres dans l'habitat individuel*.

Mission des Risques Naturels (MRN) (2021). Bilan des principaux évènements cat-clim. *Lettre d'information*, (35).

Naumann, G., Cammalleri, C., Mentaschi, L., and Feyen, L. (2021). Increased economic drought impacts in europe with anthropogenic warming. *Nature Climate Change*, 11(6):485–491.